# Dietary Fibres and the Management of Obesity and Metabolic Syndrome: The RESOLVE Study

**DOI:** 10.3390/nu12102911

**Published:** 2020-09-23

**Authors:** Angelo Tremblay, Maëlys Clinchamps, Bruno Pereira, Daniel Courteix, Bruno Lesourd, Robert Chapier, Philippe Obert, Agnes Vinet, Guillaume Walther, Elodie Chaplais, Reza Bagheri, Julien S. Baker, David Thivel, Vicky Drapeau, Frédéric Dutheil

**Affiliations:** 1Institute of Nutrition and Functional Foods (INAF), Université Laval, Quebec City, QC G1V 0A6, Canada; vicky.drapeau@fse.ulaval.ca; 2Department of Kinesiology, Université Laval, Quebec City, QC G1V 0A6, Canada; 3Preventive and Occupational Medicine, University Hospital of Clermont-Ferrand, 63000 Clermont-Ferrand, France; maelysclinchamps@gmail.com (M.C.); fred_dutheil@yahoo.fr (F.D.); 4Biostatistics, University Hospital of Clermont-Ferrand, 63000 Clermont-Ferrand, France; bpereira@chu-clermontferrand.fr; 5Laboratory of the Metabolic Adaptations to Exercise under Physiological and Pathological Conditions (AME2P), Université Clermont Auvergne, 63000 Clermont-Ferrand, France; daniel.courteix@uca.fr (D.C.); lesourd.bruno@orange.fr (B.L.); david.thivel@uca.fr (D.T.); 6Thermalia Center, 63140 Châtelguyon, France; robertchapier@yahoo.fr; 7Laboratory of Cardiovascular Pharm-ecology (LaPEC EA4278), Université d’Avignon, 84000 Avignon, France; philippe.obert@univ-avignon.fr (P.O.); agnes.vinet@univ-avignon.fr (A.V.); guillaume.walther@univ-avignon.fr (G.W.); 8Laboratory “Development, Adaption and Disability” (DevAH-EA 3450), Université de Lorraine, 54000 Nancy, France; e.chaplais@live.fr; 9Department of Exercise Physiology, University of Isfahan, Isfahan 81746-73441, Iran; will.fivb@yahoo.com; 10Centre for Health and Exercise Science Research, Department of Sport, Physical Education and Health, Hong Kong Baptist University, Kowloon Tong 999077, Hong Kong; jsbaker@hkbu.edu.hk; 11Physical Institut, Universitaire de France, 75000 Paris, France; 12Physiological and Psychosocial Stress, LaPSCo, CNRS, University Clermont Auvergne, 63000 Clermont-Ferrand, France; 13Witty Fit, 75000 Paris, France

**Keywords:** diet, exercise, body weight, macronutrient

## Abstract

Objectives: This study was performed to evaluate the long-term maintenance of nutritional changes promoted during an intensive initial intervention to induce body weight loss. The ability of these changes to predict long-term health outcomes was also examined. Methods: Nutritional variables, body composition, and metabolic markers collected in the RESOLVE project were analyzed before and after a 3-week intensive diet–exercise intervention (Phase 1), and during a subsequent supervision under free living conditions, of 12 months (Phase 2). Results: As expected, the macronutrient composition of the diet was modified to promote a negative energy balance during Phase 1. The decrease in carbohydrates imposed during this phase was maintained during Phase 2 whereas the increase in protein intake returned to baseline values at the end of the program. Dietary fiber intake was almost doubled during Phase 1 and remained significantly greater than baseline values throughout Phase 2. Moreover, fiber intake was the only nutritional variable that systematically and significantly predicted variations of health outcomes in the study. Conclusion: The adequacy of dietary fiber intake should be a matter of primary consideration in diet-based weight reduction programs.

## 1. Introduction

Numerous efforts have been deployed over the last decades to develop approaches that would permit an adequate and sustainable body weight management, especially in individuals with obesity. Obviously, the first law of thermodynamics provides the basic principle underlying this management but it is not informative about the factors which provoke the positive caloric imbalance, ultimately leading to obesity. In this regard, suboptimal macronutrient diet composition and insufficient physical activity practice have traditionally been considered as the main environmental determinants of excess body weight. Thus, even if recent research has also identified and characterized additional factors involved in the determinism of excess weight, e.g., short sleep duration [1,2] and body lipophilic pollutants [3,4], the search for optimal feeding and exercise has remained the pillar of obesity prevention and treatment.

Beyond the necessity of achieving a negative energy balance, an obesity management program must be sufficiently satiating to prevent, or at least attenuate, the decrease in satiety which progressively occurs over time with weight loss [5,6]. Since lipids have a low satiating potential [7], preventing high-fat intake, especially saturated fat, is a well-accepted feature of healthy eating [8]. Conversely, proteins are highly satiating, which probably explains the accentuation of body weight/fat loss observed in individuals with obesity who are subjected to a high protein diet [9]. Interestingly, the potency of a high protein diet to reduce body fat is increased when it is combined with an exercise program [10]. Dietary fibers are also known to influence energy intake and balance [11] and they also contribute to reduce the glycemic index of food [12,13]. This is concordant with the results obtained in the Diogenes Project, which showed that a high protein-low GI diet was a successful regimen to ensure the maintenance of a reduced body weight following caloric restriction [14]. Thus, up to now, the characteristics of a healthy diet/body weight reducing program include a low to moderate total, and saturated, lipid intake, and an increase in the relative consumption of protein and of high-fiber, low-GI carbohydrate foods. Recently, we tested the impact of a regimen integrating these features and found that it successfully accentuated body weight loss, while facilitating appetite control in individuals with a low satiety phenotype [15]. However, this type of study does not permit identifying which constituent of the macronutrient family has the greatest potential to influence energy balance in this context.

Clinical research has also showed that the strategic organization of treatment may be worth consideration, in the planning of a weight-reducing program. In this regard, the study of Barnard et al. [16] raised the question as to what extent a rigorous intervention at the beginning of a program can pave the way towards a more successful program. In non-insulin-dependent diabetes mellitus (NIDDM) diabetic patients, they found that a 26-day program based on a high complex carbohydrate, low-fat diet and exercise induced substantial changes in body weight. They also observed a pronounced decrease in fasting glycemia that was accompanied by a discontinuation of oral hypoglycemic agents or insulin therapy in a majority of participants. In the RESOLVE Project, the same approach was used in the context of a one-year intervention in overweight individuals [17]. The program was initiated with a 3-week highly supervised healthy diet aiming at a caloric deficit and a demanding exercise program, whose intensity of either endurance or resistance exercises varied between subgroups of participants. The results showed that the very significant body weight and fat loss, and the improvement of the metabolic condition, observed in this first phase of the program was maintained or even improved during the subsequent 50 weeks when participants had returned to their usual life habits [18]. Despite the numerous articles published using data from the RESOLVE study, no article focused on the influence of dietary intake on body composition (especially fat tissue), and health outcomes. In this regard, a first objective of this study was to document variations of energy and macronutrient intake during the RESOLVE project, with the objective of determining the extent to which changes promoted in Phase 1 of the project were subsequently maintained. According to the observations presented above, a second aim of the study was to determine which macronutrients predict changes in body composition and metabolic fitness in the context of the RESOLVE project.

## 2. Methods

### 2.1. Study Design

This study is a secondary analysis of the RESOLVE randomized controlled trial [18,19,20,21,22,23,24,25,26,27,28,29], where participants were randomly assigned into three groups of different intensities of exercise (same volume/time of training between groups): 1) **Re** group—high-resistance-moderate-endurance, participants performed 10 repetitions at 70% of one maximal repetition in resistance and 30% of VO2-max for endurance training, 2) **rE** group—moderate-resistance (30%)-high-endurance (70%), and 3) **re** group—moderate-resistance (30%)-moderate-endurance (30%). Randomization was computer generated with stratification according to sex, age, and body mass index. All participants followed the same restrictive diet. The study was a single blind, i.e., assessors were blinded to the participants’ group assignment. The study was approved by the ethics committee South-East I (clinicaltrials.gov NCT01354405). All participants provided written informed consent.

### 2.2. Recruitment of Participants

Participants were recruited via advertisements. Inclusion criteria were having a metabolic syndrome (MetS) according to the International Diabetes Federation (IDF) definition [30], being aged between 50 and 70 years old, a sedentary lifestyle, unchanged body weight (>2 kg), and unchanged medical treatment over the previous 6 months, no cardiovascular, hepatic, renal, endocrine, or psychiatric diseases, no use of medications that affect body weight, no restricted diet in the previous year, and a satisfactory maximal exercise tolerance test (VO_2max_) (Figure 1).

The intervention began with a 3-week residential program based on a tryptic approach (diet, exercise, education). Dietary intake was monitored by three dieticians and one medical doctor specialized in nutrition, and two master students. At least two of them were continuously on site during those three weeks. Participants received daily, both standard and personalized balanced meals, providing 1.2 g/kg/day protein, with 15 to 20% of the total energy intake from proteins, 30 to 35% from lipids, and the rest from carbohydrates. Participants were targeted to reach a negative energy balance of 500 kcal/day. Basal metabolic rate was calculated using the equations of Black [31]. For exercise training, participants were also coached daily and individually [32] and had to reach the target of their assigned group (Re, rE, or re). The same length of time was spent daily by all groups in resistance (1h30) and endurance (1h30) exercises. Groups differed only in intensity, from 30% to 70%. Patients were monitored by a Polar™ S810 with instantaneous recording and storage of heart rate values. Resistance training was done four times a week and consisted of eight exercises with free weights and traditional muscle building equipment. Each exercise was performed for three sets of 10 repetitions. Endurance training took the form of aquagym, cycling, and walking. Educational support was also a strong part of the program. Participants attended lectures on physiology of the MetS, nutrition, cooking, and exercise training. The aim of the educational support was to provide them the ability to better self-manage themselves on returning home [32].

The 3-week intervention was followed by a 1-year at-home follow-up. Participants were asked to carry out the same training program by themselves. They were seen at months 3, 6, and 12 by a dietician and a physical coach. They were also able to contact them at any time.

### 2.3. Outcomes and Parameters Measured

During the three-week residential program, food intake of participants was continuously monitored. Meals were weighed, as well as eventual food leftovers. At baseline and during follow-up, three-day self-report questionnaires on food intake and physical activity allowed the calculation of daily energy intake and daily energy expenditure. Nutrilog software was used for calculation of macro- and micro-nutrient intakes. Body composition was measured by dual-energy X-ray absorptiometry (DXA, Hologic QDR 4500 series; Waltham, MA, USA). The in vivo coefficients of variation were 4.2, 0.4, and 0.5% for fat, lean, and bone masses, respectively. Central fat, a surrogate of visceral fat, was assessed according to Kamel, et al. [33]. Fasting blood samples were drawn between 7:00 and 7:30 a.m., aliquoted and stored at −80 °C until analyses. Biological assays were performed in the biochemistry laboratory of the University Hospital (Clermont-Ferrand, France).

### 2.4. Follow-Up Assessments

All outcomes were measured at baseline (D0), at 21 days (D21), i.e., at the end of the residential program, and during the at-home follow-up at 3 months (M3), 6 months (M6), and 12 months (M12), except for food questionnaires and number of training sessions per week. They were asked every month. The outcomes were used to calculate compliance to nutrition (score from 0 to 12, i.e., 12 = 100% for food questionnaires returned) and to exercise (score from 0 to 4, i.e., 4 = 100% for the number of training sessions undertaken per week). The overall compliance score was the mean of these two scores (nutrition and physical activity).

### 2.5. Statistical Analysis

Statistical analyses were performed using Stata software (version 15, StataCorp, College Station, USA). Continuous data were expressed as means and standard-deviations (SD). The normality of the distribution was checked with a Shapiro-Wilk test. To analyze longitudinal data (variations in energy intake and macronutrient, variations in biochemical markers and blood pressure) and to evaluate the effect of fibers and nutritional factors (energy intake, protein intake, complex carbohydrate, simple carbohydrate, saturated fat, monounsaturated fat, polyunsaturated fat, cholesterol intake, fiber intake, water intake) on health outcomes (central fat, weight, BMI, fat mass, fat free mass, waist, blood glucose, HbA1c, insulimenia, Homa-IR, total cholesterol, triglycerides, HDL, LDL, Hs-CRP, SBP, DBP), marginal models, estimated using a generalized estimating equation, were used to study group (Re, rE, re) and time-point evaluation effects, and their interactions, taking into account, between and within, participant variability. The normality of residuals was checked for all models. When appropriate, a logarithmic transformation of the dependent variable was performed. Sidak’s type I error correction was applied for multiple comparisons between time-point evaluation. Multivariable analyses were then performed using the aforementioned models, with covariates chosen according to univariate results and their clinical relevance, with a particular attention paid to multicollinearity. The results were expressed as effect-sizes and 95% confidence intervals. Subgroup analyses, according to groups of different intensities of exercise, were performed. For non-repeated data, the comparisons between groups were carried out using analysis of variance (ANOVA) or non-parametric Kruskal–Wallis test, when the assumptions of ANOVA were not met. The homoscedasticity was studied using Shapiro–Wilk’s test. Two by two post-hoc comparison tests were then applied, i.e., Tukey–Kramer after ANOVA, and Dunn after Kruskal–Wallis. The relationships between continuous data were explored by estimating Pearson or Spearman correlation coefficient, with a Sidak type I error correction. Differences were considered statistically significant at *p* < 0.05.

## 3. Results

We included 100 volunteers (59.4 ± 5.1 years old, 44% men) who underwent randomization; 87 completed the whole intervention (Figure 1), without difference between participants who withdraw from the study for sociodemographic. The mean overall compliance score for both diet and exercise was 52.4 ± 22.4%, without difference between groups. There was a significant correlation (*r* = 0.72) between compliance to diet and compliance to exercise (*p* < 0.0001). Variations in energy intake and macronutrient during the two phases of the program are presented in Table 1.

As expected, a significant decrease in daily energy intake was observed during Phase 1 (D0-D21) and the achieved reduced energy intake was maintained during Phase 2 (D21-M12) for the three exercise groups. This table also shows that despite the negative energy balance imposed during Phase 1, there was a significant increase in protein intake that was however not maintained in the free living context of Phase 2. Carbohydrate intake was reduced during Phase 1, and this decrease was accentuated in Phase 2. Table 1 also shows that this reduction in carbohydrate intake was explained by changes in both complex and simple carbohydrate. A substantial decrease in total, saturated, monounsaturated, and polyunsaturated fat intake was observed during Phase 1 (Table 1). During Phase 2, all the dietary fat variables increased to an intermediate level between baseline and Phase 1 values.

The intervention of Phase 1 induced a considerable increase in dietary fiber intake which was almost doubled at D21 compared to baseline values (Table 1). This table also indicates that changes in fiber intake during Phase 1 were partly maintained when participants had recovered their usual life conditions in Phase 2. Indeed, mean fiber intake remained significantly greater at M3, M6, and M12 than at baseline (*p* < 0.01, Table 1). Water intake was also increased in Phase 1 and returned to its basal level in Phase 2.

As shown in Table 2 and as previously reported [18], the supervised diet-exercise intervention in Phase 1 induced a highly significant decrease in body weight, fat mass, fat-free mass, central fat, and waist circumference (*p* < 0.001). Except for fat-free mass, this reducing effect was accentuated to a significant extent under the free living conditions of Phase 2. It is also noteworthy to emphasize that the decrease in central fat was more pronounced in the Re and rE groups than in the re participants at every time of measurement during this protocol.

Table 2 presents variations in biochemical markers and blood pressure during the two phases of the protocol. As expected, Phase 1 improved glycaemia regulation, as reflected by the significant decrease in fasting plasma glucose and HbA1c observed at D21. Interestingly, these reduced levels remained strictly unchanged up to the end of Phase 2. Plasma total cholesterol, HDL-cholesterol, LDL-cholesterol, and TG concentrations were significantly reduced following Phase 1. Conversely, they increased during Phase 2 to such an extent that a highly significant difference (*p* < 0.001) was observed for all these variables between D21 and M12 values. The same trend was noted for systolic and diastolic blood pressure which decreased after Phase 1 and progressively returned to baseline values at the end of Phase 2. With respect to Hs-CRP, the reduction induced by the Phase 1 intervention persisted during Phase 2.

According to the second objective of the study, Figure 2 illustrates the ability to predict health outcomes from variations in nutritional factors. Additional relevant details are presented in Appendix A. As expected, there were some significant associations between variations in the intake of some nutritional variables and health outcomes. However, fiber intake was the only nutritional variable which systematically predicted a significant reduction in health indicators except for fat-free mass, HDL-cholesterol, and systolic blood pressure.

We further examined the relationship between fiber intake and health outcomes by evaluating these associations within each exercise group. As shown in Figure 3, body composition indicators, except for fat-free mass, were favorably related to fiber intake, especially in the rE group. Interestingly, this apparent beneficial effect increased over time.

## 4. Discussion

This study was realized in the context of a global obesity epidemic, which requires the validation of novel approaches to facilitate the management of excess body fat. As explained above, we studied the approach used in the RESOLVE project, which is known for its beneficial effects on body composition [18] and metabolic health [17]. In brief, the RESOLVE protocol was initiated by an intensive intervention of several weeks that was followed by a second phase of distance supervision during which participants were instructed to maintain the lifestyle changes promoted during the first phase of the study. No articles from the RESOLVE trial have to date investigated the influence of nutritional aspects. The specific aims of the present study were to evaluate the long term maintenance of dietary changes induced in Phase 1 of the project and to determine the extent to which these dietary changes can predict the impact of the program on body composition and health-related metabolic indicators.

The dietary modifications that were imposed during Phase 1 were concordant with recognized good practices, favoring a decrease in energy intake while preserving an adequate satiating potential of the diet. The reduced energy intake imposed on participants during Phase 1 was attributable to a decrease in carbohydrate and lipid intake. Conversely, protein intake was increased, which agrees with the previous demonstration that a high-protein diet favors an accentuation of body weight/fat loss [9] and a better weight loss maintenance in a weight-reduced obese state [14]. It is also relevant to emphasize that the substantial increase in dietary fiber that occurred in Phase 1 is also known to facilitate appetite control and weight loss [11]. In this regard, the first objective of this study was to evaluate the maintenance over time of these nutritional changes. The results revealed a maximal variation in changes in macronutrient intake following Phase 1. Indeed, the absolute and relative decrease in carbohydrate observed in Phase 1 was strictly maintained to the end of the study. However, the increase in protein intake that was achieved under the supervised conditions of Phase 1 had totally disappeared at the end of Phase 2. Intermediate changes occurred for lipid intake since it increased to a mid-level between baseline and the end of Phase 1, when subjects had recovered the free living context of Phase 2. Fiber intake also partly returned towards baseline values after Phase 1, although values recorded at each time of measurement in Phase 2 were significantly greater than baseline intake. Globally, the changes in macronutrient intake during Phase 2 suggest that the gain in the satiating potential of the diet that was achieved in Phase 1 was at least partly preserved in Phase 2. This interpretation is concordant with the observation that body weight and fat continued to decrease in Phase 2.

The clinical significance of changes in macronutrient intake also represented a major preoccupation in this study. Thus, the second objective of the study was to assess the association between these changes and those in health outcomes that included markers of body composition, metabolic health, and systemic inflammation. Interestingly, dietary fiber was the only nutritional factor that significantly and systematically predicted variations in health outcomes. The explanation of this association is not clear, although it seems logical to make a link between the fact that fiber intake was the most influenced nutritional variable during Phase 1. Moreover, this favorable change persisted at a sufficiently high level in Phase 2 to remain significantly greater than baseline intake. Another potential explanation of the relationship between dietary fiber and the health outcomes of this study is the ability of fiber to modify the gut microbiota [34,35]. If this is the case, further research will be needed to provide a specific characterization of this hypothesis.

We have recently reported that exercise modalities in the RESOLVE study influenced the response of body composition to the program (18). In this regard, it thus appeared relevant to determine if a fiber–exercise relationship might be preferentially detected for a specific modality of exercise practice. As illustrated in Figure 3, this association was particularly obvious in the rE group which was exposed to a more demanding aerobic exercise program. Although this observation is compatible with the notion that diet composition influences the response to exercise, further research will be needed to confirm that body fat loss is influenced by the optimal combination of fiber intake and vigorous aerobic exercise. Furthermore, it will also be relevant to establish if this represents an interaction effect or the summation of two independent effects.

Even if the present study was not primarily undertaken to document the body fat/metabolic health relationship, it is relevant to emphasize that results reported here provide a good example of the non-systematic association between these indicators. For instance, once Phase 1 was completed, body weight and fat, as well as central fat, continued to decrease whereas plasma lipid-lipoprotein concentrations and blood pressure were significantly increased.

The present study has some strengths and limitations related to the measurement of dietary intake, which ultimately complement each other to support the validity of reported results. Indeed, the use of self-reported recalls at baseline and during Phase 2 might have led to some underreported values. On the other hand, the precise measurements of energy and nutrient intake in Phase 1, as well as body composition determinations, are useful to estimate potential food intake underreporting. Indeed, Table 1 shows that the mean decrease in daily energy intake during Phase 1 was 330 kcal whereas the personalized regimen of participants was targeting a decrease of 500 kcal/day during this phase. It is also relevant to consider the energy equivalent of morphological changes, and the energy cost of the exercise program during Phase 1, to have an adequate perception of the validity of food intake measurements in the present study. As shown in Table 2, the Phase 1 intervention induced a mean decrease in fat mass and fat-free mass of 2.7 and 0.9 kg, respectively. This corresponds to an energy equivalent of 25,500 kcal which represents a daily energy deficit of 1200 kcal/day and thus leaves a gap of about 970 kcal/day between the estimated deficit and the decrease in energy intake during this phase. As previously reported [18], participants had to perform exercise for 15–20 h/week during Phase 1. If we assume that exercise induced an excess energy expenditure of about 6 kcal/min, this means that the exercise regimen imposed a supplementary energy expenditure of 750 to 1000 kcal/day. Thus, the additional energy cost of exercise can explain a major part of the gap between the energy equivalent of weight loss and the decrease in energy intake during Phase 1, although it remains possible that dietary recalls might have underestimated energy intake by several hundred kcal/day. However, the stability of measurements performed at baseline versus Phase 2, gives confidence in the analyses performed in this study. The mean compliance to diet and exercise guidelines was adequate in the present study. However, we want to acknowledge that maintaining such a level of high compliance was costly, and therefore the study may lack generalisability as a public health strategy within the community.

In summary, this study shows that the favorable nutritional changes imposed during the initial intensive intervention of the RESOLVE Project were partly maintained when participants were followed for almost a year, when they had recovered their free living context. This observation is particularly valid for dietary fiber intake which almost doubled during Phase 1, and remained significantly higher than baseline intake during the Phase 2 follow ups. Interestingly, dietary fiber appeared as the main predictor of variation in the health outcomes of the study. This suggests that more attention should be given to the adequacy of fiber intake in programs aiming at the improvement of body composition and metabolic health.

## Figures and Tables

**Figure 1 nutrients-12-02911-f001:**
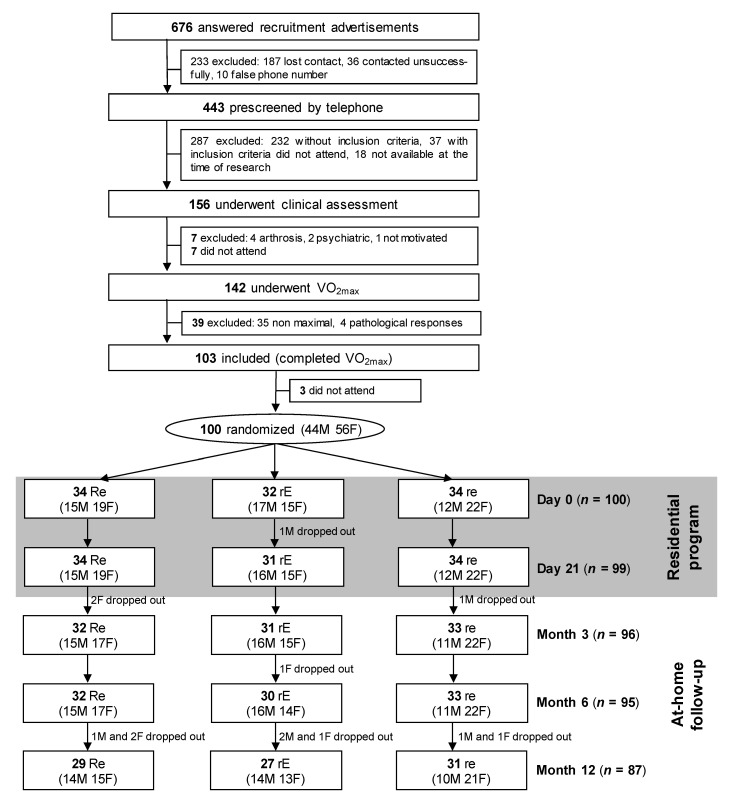
Flow chart. VO_2max_ = maximal oxygen uptake; **Re, rE** and **re:** see methods for details; M= males; F = females.

**Figure 2 nutrients-12-02911-f002:**
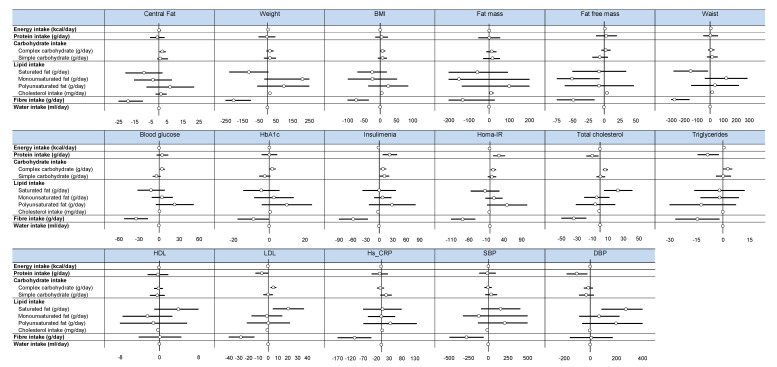
Effect of nutritional factors on health outcomes—Forest plots represent effect sizes (dot) and their 95% confidence intervals (horizontal lines on both sides). For details, please see Appendix A. Data are adjusted on age. The effect of each variable on the outcome is represented in the forest-plot by a dot on a horizontal line. The black dots represent the coefficient for each variable, and the length of each line around the dots represent its 95% confidence interval (95CI). The black solid vertical line represents the null estimate (with a value of 0). Horizontal lines that cross the null vertical line represent non-significant variables on the outcome. HbA1c: glycated hemoglobin; Homa-IR: Homeostatic Model Assessment of Insulin Resistance, HDL: high-density lipoprotein, LDL: low-density lipoprotein, Hs-CRP: high-sensitivity C—reactive protein. BMI: Body Mass Index. SBP: systolic blood pressure; DBP: diastolic blood pressure.

**Figure 3 nutrients-12-02911-f003:**
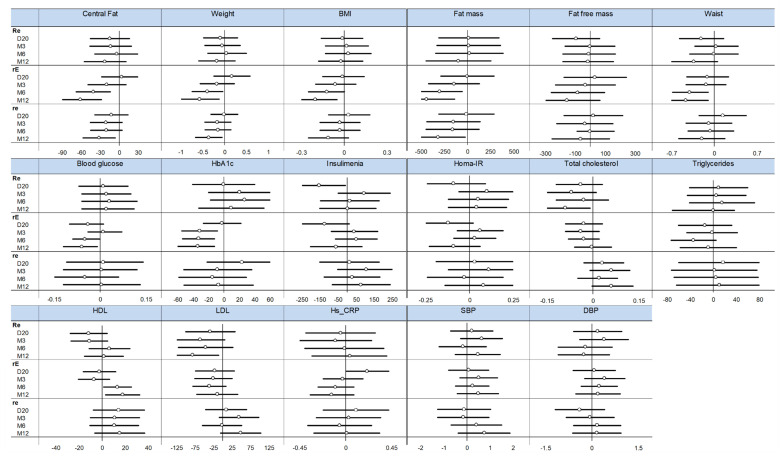
Effect of fiber intake on health outcomes depending on time and group—Forest plots represent effect sizes (dot) and their 95% confidence intervals (horizontal lines on both sides). For details, please see Appendix A. HbA1c: glycated haemoglobin; Homa-IR: Homeostatic Model Assessment of Insulin Resistance, HDL: high-density lipoprotein, LDL: low-density lipoprotein, Hs-CRP: high-sensitivity c-reactive protein. BMI: Body Mass Index. SBP: systolic blood pressure; DBP: diastolic blood pressure. Re, rE and re: see methods for details. D: day; M: month.

**Table 1 nutrients-12-02911-t001:** Energy and nutrient intake during the program.

Outcomes	Changes from Baseline				Time	Group	Interaction	Between Groups	Re	rE	Re
Variables	All	Re	rE	re	Effect	Effect		Differences	vs re	vs re	vs rE
**Energy Intake (kcal/day)**					**0.003**	0.233	0.499		0.490	0.341	0.769
D0	1821 ± 499	1894 ± 460	1893 ± 619	1684 ± 391							
D21	1574 ± 117	1583 ± 96	1624 ± 91	1521 ± 135	***		0.195	Re > re	**0.024**	0.239	0.528
M3	1599 ± 275	1655 ± 320	1585 ± 250	1551 ± 239	*		**0.046**	Re = rE = re	0.095	0.059	0.570
M6	1623 ± 257	1670 ± 265	1693 ± 287	1516 ± 186	**		0.647	Re = rE = re	0.315	0.683	0.704
M12	1591 ± 226	1632 ± 249	1631 ± 226	1520.0 ± 187	***		0.164	Re = rE = re	0.141	0.199	0.868
**Protein Intake (g/day)**					**<0.001**	0.112	0.339		0.640	0.775	0.467
D0	80.1 ± 20.3	81 ± 17.8	85.8 ± 25.1	74.2 ± 16.6							
D21	102.7 ± 9.9	102.4 ± 8.2	106.2 ± 9.2	99.9 ± 11.4	***/^‡‡‡^/^###^/^¥¥¥^		0.178	Re = rE = re	0.196	0.220	0.875
M3	83.1 ± 13.5	82.8 ± 12.7	85.2 ± 14.5	81.7 ± 13.8	**/^†††^		0.052	Re = rE = re	0.085	0.076	0.706
M6	83.3 ± 15.0	83.3 ± 11.8	91.7 ± 12.0	76.2 ± 16.5	^†††^		0.433	Re = rE = re	0.893	0.451	0.344
M12	81.9 ± 11.1	80.6 ± 11.0	88.3 ± 9.6	78.4 ± 10.4	^†††^		0.383	Re = rE = re	0.179	0.466	0.710
**Carbohydrate Intake (g/day)**					**<0.001**	0.237	0.932		0.821	0.959	0.870
D0	191.8 ± 60.2	200.0 ± 61.8	196.8 ± 69.5	178.9 ± 47.8							
D21	174.3 ± 20.5	176.8 ± 19.3	181.6 ± 19.0	165.5 ± 20.4	*		0.868	Re = rE = re	0.327	0.884	0.470
M3	171.6 ± 30.0	178.9 ± 33.0	173.4 ± 30.2	162.3 ± 24.6	**		0.649	Re = rE = re	0.615	0.618	0.990
M6	169.1 ± 31.1	175.3 ± 30.5	180.7 ± 33.1	152.9 ± 23.2	***		0.458	Re = rE = re	0.857	0.406	0.397
M12	165.9 ± 39.5	176.4 ± 55.5	167.9 ± 29.7	154.3 ± 21.1	***		0.775	Re = rE = re	0.943	0.707	0.806
**Carbohydrate Intake (% kcal)**					**<0.05**	0.175	0.133		0.866	**0.015**	**0.010**
D0	44.4 ± 5.7	45.6 ± 5.5	43.7 ± 6.5	43.8 ± 5.3							
D21	44.4 ± 3.1	44.8 ± 3.2	44.8 ± 3.5	43.7 ± 2.7			0.634	Re = rE = re	0.727	0.676	0.544
M3	44.8 ± 4.4	45.4 ± 4	45.8 ± 4.8	43.2 ± 4.2	^¥^		0.285	Re = rE = re	0.510	0.363	0.468
M6	43 ± 4.6	43.6 ± 4.7	44 ± 4.5	41.5 ± 4.4	*		0.317	Re = rE = re	0.747	0.390	0.460
M12	43.5 ± 14.3	42.3 ± 3.2	47.7 ± 26.6	41.5 ± 3.3	**/^†^		**0.018**	rE > Re = re	0.488	**0.036**	**0.016**
**Complex Carbohydrate (g/day)**					**<0.001**	0.382	0.987		0.757	0.851	0.631
D0	128.6 ± 48.5	132.9 ± 46.0	134.9 ± 55.9	118.5 ± 43.4							
D21	110.9 ± 19.7	113.1 ± 18.3	118.9 ± 18.1	101.6 ± 19.2	***		0.892	Re = rE = re	0.739	0.855	0.663
M3	112.4 ± 23.9	114.8 ± 24.1	116.9 ± 25.7	105.9 ± 21.4	*		0.608	Re = rE = re	0.543	0.571	0.934
M6	109.3 ± 24.0	113.8 ± 24.2	118.5 ± 24.4	96.8 ± 18.0	***		0.684	Re = rE = re	0.931	0.660	0.755
M12	109.3 ± 36.4	116.5 ± 52.0	113.9 ± 24.9	98.9 ± 20.3	***		0.828	Re = rE = re	0.826	0.728	0.667
**Simple Carbohydrate (g/day)**					**0.012**	0.134	0.101		0.439	0.585	0.202
D0	65.2 ± 24.9	72.7 ± 30.7	61.9 ± 22.3	60.4 ± 18.5							
D21	63.5 ± 8.3	63.7 ± 7.8	62.7 ± 9.9	63.8 ± 7.3	^‡^/^#^/^¥¥^		0.468	re = rE > Re	**0.003**	0.507	**0.043**
M3	59.3 ± 15.0	64.1 ± 15.0	56.6 ± 17.5	56.4 ± 11.5	^†^		0.942	Re = rE = re	0.420	0.917	0.507
M6	59.8 ± 14.8	61.5 ± 13.3	62.2 ± 14.5	56.1 ± 16.2	^†^		0.349	rE > Re	0.140	0.293	**0.022**
M12	56.6 ± 11.2	59.9 ± 11.6	54.1 ± 10.8	55.4 ± 10.7	^††^		0.811	Re = rE = re	0.128	0.779	0.257
**Lipid Intake (g/day)**					**<0.001**	0.365	0.128		0.394	0.101	0.409
D0	72.9 ± 24.9	75.8 ± 26.3	76.4 ± 29.3	66.6 ± 17.5							
D21	51.3 ± 4.7	51.3 ± 4.5	52.0 ± 4.7	50.7 ± 4.8	***/^‡‡‡^/^###^/^¥¥¥^		0.071	Re < re	**0.028**	0.077	0.884
M3	58.5 ± 14.9	60.4 ± 17.1	54.3 ± 12.7	60.1 ± 13.7	*/^†††^		**0.001**	Re = rE > re	**0.030**	**0.001**	0.213
M6	62.8 ± 14.2	64.2 ± 14.6	62.3 ± 16.5	61.8 ± 11.8	^†††^		0.056	Re = rE = re	0.133	0.058	0.587
M12	63.4 ± 10.8	65.0 ± 12.1	63.7 ± 11.8	61.7 ± 8.5	^†††^		0.066	Re = rE = re	0.171	0.075	0.526
**Lipid Intake (% kcal)**					**<0.001**	0.829	0.239		0.946	0.105	0.093
D0	36.8 ± 5.5	36.3 ± 5.2	37.0 ± 6.9	37.1 ± 4.2							
D21	29.7 ± 2.6	29.5 ± 2.6	29.2 ± 2.2	30.3 ± 2.7	***/^‡‡‡^/^###^/^¥¥¥^		0.455	Re = rE = re	0.846	0.467	0.567
M3	33.6 ± 5.2	33.5 ± 4.8	31.9 ± 5.7	35.4 ± 4.8	*/^†††^		**0.031**	rE < re	0.390	**0.036**	0.163
M6	35.4 ± 5.6	35.4 ± 5.3	33.4 ± 6.4	37.1 ± 4.8	^†††^		**0.016**	rE < re	0.526	**0.019**	0.077
M12	36.5 ± 3.9	36.8 ± 3.8	35.5 ± 4.5	37.0 ± 3.4	^†††^		0.243	Re = rE = re	0.720	0.236	0.146
**Saturated Fat (g/day)**					**<0.001**	0.844	0.630		0.862	0.538	0.435
D0	30.4 ± 11.1	31.1 ± 12	31.0 ± 13.2	29.0 ± 8.0							
D21	20.5 ± 2.1	20.4 ± 1.8	21.1 ± 2.2	20.2 ± 2.1	***/^‡‡‡^/^###^/^¥¥¥^		0.617	Re = rE = re	0.282	0.637	0.668
M3	24.5 ± 7.1	26.1 ± 8.5	22.2 ± 5.5	24.8 ± 6.2	***/^†††^		**0.050**	rE < re	0.573	**0.045**	0.156
M6	26.4 ± 6.5	27.2 ± 6.9	26.6 ± 7.7	25.5 ± 5.0	**/^†††^		0.714	Re = rE = re	0.841	0.698	0.854
M12	26.2 ± 5.4	27.3 ± 5.7	26.1 ± 6.3	25.1 ± 4.0	**/^†††^		0.540	Re = rE = re	0.932	0.546	0.581
**Monounsaturated Fat (g/day)**					**<0.001**	0.428	**0.008**		0.360	**0.012**	0.100
D0	29.3 ± 10.3	29.6 ± 9.6	32.2 ± 12.7	26.3 ± 7.8							
D21	21.7 ± 2.7	21.8 ± 2.7	21.5 ± 2.8	21.8 ± 2.6	***/^‡‡^/^###^/^¥¥¥^		**0.003**	Re = rE < re	**0.044**	**0.004**	0.234
M3	24.1 ± 6.6	24.4 ± 7.3	22.3 ± 6.1	25.3 ± 6.2	^††^		**0.000**	rE < Re < re	**0.011**	**0.000**	**0.043**
M6	26.1 ± 6.5	26.6 ± 6.5	25.5 ± 7.5	26.2 ± 5.7	^†††^		**0.002**	rE << re	0.087	**0.002**	0.108
M12	26.9 ± 4.7	27.4 ± 5.1	26.9 ± 5.1	26.3 ± 4.2	^†††^		**0.006**	rE >> re	0.183	**0.008**	0.114
**Polyunsaturated Fat (g/day)**					**0.012**	0.513	0.168		0.288	0.147	0.662
D0	12.4 ± 4.9	12.8 ± 5.4	13.2 ± 5.3	11.3 ± 3.6							
D21	9.0 ± 1.6	9.0 ± 1.7	9.3 ± 1.8	8.7 ± 1.4	**/^‡‡^		0.265	Re = rE = re	0.329	0.288	0.893
M3	10.3 ± 4.7	9.9 ± 2.5	9.8 ± 2.6	11.3 ± 7.2	^††^		**0.002**	Re = rE << re	**0.009**	**0.004**	0.626
M6	10.3 ± 2.1	10.5 ± 2.1	10.3 ± 2.3	10.0 ± 2.1			0.137	Re = rE = re	0.375	0.152	0.513
M12	10.4 ± 1.9	10.3 ± 2.1	10.8 ± 1.6	10.2 ± 1.9			0.165	Re = rE = re	0.234	0.187	0.824
**Cholesterol Intake (mg/day)**					**<0.001**	**0.036**	0.476		0.437	0.944	0.504
D0	302.0 ± 112.1	289.1 ± 91.2	330.7 ± 138.6	289.1 ± 103.0							
D21	326.5 ± 79.8	327.6 ± 75.5	318.5 ± 87.6	332.3 ± 78.7	*/^‡‡‡^/^###^/^¥¥¥^		**0.022**	Re = re < rE	0.780	**0.033**	**0.034**
M3	257.1 ± 67.1	260.9 ± 78.8	253.5 ± 57.6	256.2 ± 63.2	^†††^		0.132	Re = rE = re	0.985	0.147	0.134
M6	270.4 ± 68.4	271.6 ± 66.1	280.9 ± 61.9	260.2 ± 76.4	^†††^		0.546	Re = rE = re	0.650	0.540	0.307
M12	275.9 ± 58.1	277.4 ± 50.0	293.0 ± 77.4	261.6 ± 45.5	^†††^		0.582	Re = rE = re	0.552	0.608	0.254
**Fiber Intake (g/day)**					**<0.001**	**0.004**	0.127		0.458	0.754	0.695
D0	16.7 ± 6.4	17.4 ± 4.2	18.8 ± 9.5	14.1 ± 3.7							
D21	30.7 ± 4.6	31.4 ± 5.0	31.2 ± 4.4	29.5 ± 4.3	***/^‡‡‡^/^###^/^¥¥¥^		**0.033**	rE > re	0.237	**0.043**	0.283
M3	21.6 ± 4.9	22.1 ± 4.9	21.6 ± 5.0	21.2 ± 5.0	***/^†††^		**0.003**	Re = rE > re	**0.043**	**0.004**	0.212
M6	22.7 ± 5.2	22.4 ± 4.0	24.9 ± 5.6	21.1 ± 5.5	***/^†††^		0.400	Re = rE = re	0.071	0.457	0.472
M12	21.5 ± 4.6	21.8 ± 4.1	22.9 ± 4.6	20.2 ± 4.7	**/^†††^		0.105	Re = rE = re	0.163	0.129	0.683
**Water Intake (mL/day)**					**<0.001**	**0.036**	**0.050**		0.342	0.837	0.488
D0	2060 ± 708	2190 ± 727	2170 ± 777	1832 ± 575							
D21	2995 ± 525	2907 ± 375	3095 ± 641	2995 ± 540	***/^‡‡‡^/^###^/^¥¥¥^		0.094	Re < re	**0.001**	0.088	0.203
M3	2309 ± 706	2229 ± 580	2518 ± 806	2214 ± 722	***/^†††^		0.999	re = rE > Re	**0.016**	0.889	**0.046**
M6	2171 ± 599	2108 ± 508	2390 ± 776	2043 ± 458	***/^†††^		0.712	re = rE > Re	**0.012**	0.749	0.080
M12	2149 ± 495	2151 ± 500	2255 ± 528	2069 ± 465	**/^†††^		0.204	Re > re	**0.044**	0.175	0.646

Re, rE and re: see methods for details. Bold for significant difference. D: day; M: month.*: *p* < 0.05 vs. D0; **: *p* < 0.01 vs. D0; ***: *p* < 0.001 vs. D0; ^†^: *p* < 0.05 vs. D20; ^††^: *p* < 0.01 vs. D20; ^†††^: *p* < 0.001 vs. D20; ^‡^: *p* < 0.05 vs. M3; ^‡‡^: *p* < 0.01 vs. M3; ^‡‡‡^: *p* < 0.001 vs. M3; ^#^: *p* < 0.05 vs. M6; ^###^: *p* < 0.001 vs. M6; ^¥^: *p* < 0.05 vs. M12; ^¥¥^: *p* < 0.01 vs. M12; ^¥¥¥^: *p* < 0.001 vs. M12.

**Table 2 nutrients-12-02911-t002:** Health outcomes evolution.

Outcomes	Changes from Baseline				Time	Group	Interac-	Between Groups	Re	rE	Re
Variables	All	Re	rE	re	Effect	Effect	tion	Differences	vs re	vs re	vs rE
**BMI**					**<0.001**	**<0.05**	**0.044**		0.491	**0.006**	**0.001**
D0	33.4 ± 4.1	32.1 ± 3.9	34.4 ± 4.2	33.9 ± 4							
D21	32.2 ± 3.9	30.8 ± 3.8	33 ± 3.9	32.7 ± 3.8	***/^‡‡‡^/^###^/^¥¥¥^		0.472	rE = re = Re	0.760	0.501	0.651
M3	30.9 ± 3.8	29.6 ± 3.7	31.5 ± 3.6	31.6 ± 3.9	***/^†††^/^¥^		0.084	re = rE = Re	0.887	0.103	0.127
M6	31.1 ± 4.0	30.2 ± 4.1	31.5 ± 4	31.7 ± 3.9	***/^†††^		**0.034**	Re = re > rE	0.436	**0.043**	**0.007**
M12	31.0 ± 4.0	29.9 ± 3.9	31.3 ± 4	31.8 ± 4	***/^†††^/^‡^		**0.015**	Re = re > rE	0.496	**0.019**	**0.004**
**Weight (g/day)**					**<0.001**	**0.0547**	**0.027**		0.584	0.211	0.078
D0	89.3 ± 13.3	85.4 ± 12.4	94 ± 13.7	89 ± 12.7							
D21	84.8 ± 12.4	81.9 ± 11.7	87.1 ± 13.1	85.7 ± 12.1	***/^‡‡‡^/^###^/^¥¥^		**0.000**	rE > re = Re	0.757	**0.001**	**0.002**
M3	82.6 ± 12.1	79.1 ± 11.3	86.1 ± 12.3	82.6 ± 12.2	***/^†††^		0.144	rE = re = Re	0.820	0.163	0.115
M6	83.0 ± 13.0	80.4 ± 12.6	86.1 ± 13.6	82.8 ± 12.5	***/^†††^		**0.040**	rE > re = Re	0.407	**0.048**	**0.008**
M12	82.2 ± 12.6	79.2 ± 11.9	84.9 ± 12.9	82.5 ± 12.7	***/^††^		**0.046**	rE > re = Re	0.671	0.055	**0.025**
**Fat Mass (kg)**					**<0.001**	**0.0214**	**0.035**		0.617	**0.011**	**0.003**
D0	30.7 ± 7.9	27.7 ± 7.6	32.2 ± 7.7	32.3 ± 7.6							
D21	28.0 ± 7.6	24.9 ± 7.1	29.7 ± 7.3	30.1 ± 7.4	***/^‡‡‡^/^###^/^¥¥^		0.268	re = rE= Re	0.246	0.305	0.905
M3	25.7 ± 7.2	22.1 ± 6.9	26.3 ± 6.8	28.3 ± 6.8	***/^†††^		**0.033**	re > rE	0.592	**0.044**	0.112
M6	25.7 ± 7.7	23.2 ± 8.3	26 ± 7.6	28 ± 6.7	***/^†††^		**0.008**	Re = re > rE	0.760	**0.012**	**0.005**
M12	26.1 ± 7.8	22.7 ± 7	26.7 ± 8.1	28.5 ± 7.3	***/^††^		**0.032**	Re = re > rE	0.773	**0.038**	**0.021**
**Fat Free Mass (kg)**					**<0.001**	0.1462	0.068		0.698	**0.004**	**0.001**
D0	58.5 ± 11.1	57.5 ± 10.8	61.8 ± 11.8	56.5 ± 10.7							
D21	57.6 ± 10.5	56.9 ± 10.2	60.7 ± 10.8	55.6 ± 10.3	***/^‡^		0.811	rE = Re = re	0.331	0.792	0.270
M3	56.8 ± 10.9	56.5 ± 10.6	59.5 ± 10.9	54.5 ± 10.9	***/^†^		0.241	rE = Re = re	0.935	0.246	0.227
M6	57.3 ± 10.7	57.1 ± 10.4	60.3 ± 10.8	54.8 ± 10.6	***		0.214	rE = Re = re	0.443	0.212	0.057
M12	56.3 ± 10.8	56.8 ± 11.1	58.1 ± 10.4	54.2 ± 10.9	***		**0.006**	rE >> Re = re	0.530	**0.006**	**0.001**
**Central Fat (g)**					**<0.001**	0.3123	**0.011**		0.262	**0.004**	0.087
D0	3051 ± 754	2900 ± 695	3184 ± 774	3082 ± 788							
D21	2667 ± 675	2387 ± 582	2789 ± 678	2837 ± 688	***/^‡‡‡^/^###^/^¥¥¥^		0.076	re > rE > Re	**0.000**	0.088	0.213
M3	2415 ± 690	2124 ± 560	2505 ± 732	2599 ± 690	***/^†††^		**0.028**	re > rE > Re	**0.012**	**0.030**	0.952
M6	2370 ± 761	2178 ± 733	2378 ± 819	2550 ± 709	***/^†††^/^¥¥¥^		**0.002**	re > rE > Re	**0.041**	**0.002**	0.224
M12	2447 ± 725	2213 ± 629	2451 ± 731	2656 ± 757	***/^†††/###^		**0.006**	re > rE > Re	**0.038**	**0.006**	0.387
**Waist Circumference (cm)**					**<0.001**	0.1142	**0.019**		0.331	**0.000**	**0.010**
D0	102.2 ± 10	99.6 ± 9.1	105.3 ± 11.2	102.1 ± 9.2							
D21	97.2 ± 8.9	94.5 ± 8.9	99.2 ± 9.3	97.9 ± 8.2	***/^‡‡‡^/^#^/^¥¥^		0.144	rE = re = Re	0.418	0.139	0.495
M3	94.0 ± 8.8	90.9 ± 7.7	95.9 ± 9.8	95.3 ± 8.3	***/^†††^		**0.040**	rE > re	0.095	**0.040**	0.621
M6	94.4 ± 9.5	92.3 ± 9.2	95.0 ± 10.6	96 ± 8.5	***/^†^		**0.001**	Re = re > rE	0.305	**0.001**	**0.028**
M12	93.9 ± 8.6	91.7 ± 8.0	94.4 ± 9.1	95.6 ± 8.7	***/^††^		**0.000**	Re = re > rE	0.226	**0.000**	**0.025**
**Blood Glucose**					**<0.001**	0.7996	**0.280**		0.973	0.174	0.184
D0	5.5 ± 1.4	5.5 ± 1.2	5.2 ± 1.0	5.7 ± 2							
D21	4.7 ± 0.9	4.7 ± 1.1	4.7 ± 0.6	4.8 ± 0.9	***		0.110	rE = re = Re	0.561	0.115	0.262
M3	4.6 ± 0.9	4.6 ± 0.8	4.7 ± 1.0	4.5 ± 1.0	***		**0.010**	rE = re	0.257	**0.011**	0.116
M6	4.8 ± 1.0	4.8 ± 0.9	4.7 ± 0.7	4.8 ± 1.3	***		0.146	rE = re = Re	0.329	0.156	0.638
M12	4.8 ± 0.8	4.6 ± 0.5	4.9 ± 1.0	4.9 ± 1.0	***		0.087	rE > Re	0.873	0.085	**0.030**
**HbA1c**					**<0.001**	0.9863	**0.704**		0.164	0.865	0.237
D0	6.3 ± 0.8	6.3 ± 0.8	6.2 ± 0.6	6.4 ± 0.9							
D21	6.0 ± 0.7	6.0 ± 0.7	6.0 ± 0.5	6.1 ± 0.8	***		0.968	rE = re = Re	0.717	0.978	0.729
M3	5.9 ± 0.5	5.9 ± 0.5	5.9 ± 0.4	6.0 ± 0.7	***		0.733	rE = re = Re	0.434	0.737	0.612
M6	5.8 ± 0.5	5.8 ± 0.5	5.8 ± 0.4	5.9 ± 0.7	***		0.562	rE = re = Re	0.756	0.545	0.794
M12	6.0 ± 0.6	5.9 ± 0.5	6.0 ± 0.5	6.1 ± 0.8	***		0.913	rE = re = Re	0.219	0.865	0.139
**Insulinemia (UI/L)**					**<0.001**	0.1183	**0.231**		0.160	0.064	0.629
D0											
D21	3884 ± 1379	3593 ± 1261	3701 ± 1281	4348 ± 1495	^###^		0.208	rE = re = Re	0.294	0.213	0.801
M3	3655 ± 1724	3486 ± 1977	3717 ± 1713	3766 ± 1485	^###^		0.004	rE = Re < re	**0.036**	**0.005**	0.392
M6	4196 ± 1740	4098 ± 1727	4621 ± 1852	3906 ± 1620	**/^†††/‡‡‡^/^¥¥¥^		0.301	rE = re = Re	0.223	0.311	0.895
M12	5053 ± 1454	4969 ± 1296	5012 ± 1604	5173 ± 1496	^###^		0.022	rE < re	0.101	**0.024**	0.470
**Homa-IR**					**<0.001**	0.2460	0.062		0.204	**0.040**	0.415
D0	3.9 ± 2.1	3.6 ± 1.6	3.5 ± 1.4	4.6 ± 2.8							
D21	3.1 ± 1.6	3.0 ± 1.9	3.1 ± 1.5	3.2 ± 1.4	***/^##^		0.050	rE = re = Re	0.138	0.060	0.615
M3	3.6 ± 2.1	3.5 ± 1.9	4.1 ± 2.2	3.3 ± 2.1	**/^##^		0.001	rE = Re >>> re	**0.023**	**0.001**	0.219
M6	4.4 ± 1.9	4.4 ± 1.7	4.2 ± 1.6	4.7 ± 2.4	^††^/^‡‡^/^¥¥^		0.205	rE = re = Re	0.173	0.228	0.912
M12	3.6 ± 1.7	3.3 ± 1.4	3.9 ± 1.8	3.5 ± 1.8	**/^##^		0.006	rE > re	0.124	**0.008**	0.190
**Total Cholesterol**					**<0.001**	0.7629	**0.263**		0.444	0.889	0.376
D0	5.5 ± 1.0	5.6 ± 1.2	5.5 ± 0.8	5.4 ± 0.8							
D21	4.2 ± 0.9	4.2 ± 1.0	4.3 ± 0.9	4.2 ± 0.8	***/^‡‡‡^/^###^/^¥¥¥^		0.837	rE = re = Re	0.334	0.842	0.437
M3	5.3 ± 1.1	5.4 ± 1.2	5.4 ± 1.1	5.1 ± 0.9	*/^†††^/^###^/^¥¥¥^		0.206	rE = re = Re	0.404	0.178	0.676
M6	5.3 ± 1.0	5.3 ± 1.1	5.3 ± 0.8	5.3 ± 1.0	^†††^		0.714	rE = re = Re	0.371	0.705	0.587
M12	5.3 ± 0.9	5.5 ± 1.1	5.3 ± 0.7	5.2 ± 0.8	^†††^		1.00	rE = re = Re	0.296	0.950	0.285
**Triglycerides**					**<0.001**	0.5708	0.650		0.723	0.413	0.636
D0	1.9 ± 0.9	1.8 ± 0.6	1.8 ± 0.9	2.2 ± 1.2							
D21	1.3 ± 0.4	1.3 ± 0.4	1.3 ± 0.3	1.3 ± 0.3	***/^##^/^¥¥¥^		0.134	rE = re = Re	0.089	0.174	0.906
M3	1.6 ± 0.7	1.4 ± 0.4	1.6 ± 0.9	1.6 ± 0.7	***		0.046	rE = re = Re	0.320	0.065	0.257
M6	1.7 ± 0.9	1.6 ± 0.7	1.7 ± 1.2	1.8 ± 0.9	**/^††^		0.228	rE = re = Re	0.296	0.269	0.814
M12	1.8 ± 0.9	1.6 ± 0.7	1.8 ± 1.2	1.9 ± 1.0	*/^†††^		0.237	rE = re = Re	0.433	0.275	0.629
**HDL Cholesterol**					**<0.001**	0.9121	0.412		0.859	0.296	0.384
D0	1.2 ± 0.3	1.2 ± 0.2	1.2 ± 0.2	1.2 ± 0.3							
D21	1.1 ± 0.3	1.2 ± 0.2	1.1 ± 0.2	1.2 ± 0.3	^##^/^¥¥¥^		0.395	rE = re = Re	0.893	0.402	0.291
M3	1.3 ± 0.3	1.3 ± 0.3	1.2 ± 0.3	1.3 ± 0.4			0.625	rE = re = Re	0.214	0.605	0.067
M6	1.3 ± 0.3	1.3 ± 0.3	1.3 ± 0.3	1.3 ± 0.4	**/^††^		0.955	rE = re = Re	0.741	0.948	0.687
M12	1.3 ± 0.3	1.3 ± 0.3	1.3 ± 0.3	1.3 ± 0.4	**/^†††^		0.458	rE = re = Re	0.661	0.475	0.743
**LDL Cholesterol**					**<0.001**	0.2260	0.081		0.403	0.717	0.239
D0	3.5 ± 0.9	3.6 ± 1.1	3.5 ± 0.8	3.3 ± 0.8							
D21	2.5 ± 0.8	2.4 ± 0.9	2.6 ± 0.8	2.5 ± 0.8	***/^‡‡‡^/^###^/^¥¥¥^		0.633	rE = re = Re	0.108	0.629	0.269
M3	3.3 ± 1.0	3.4 ± 1.1	3.5 ± 1	3.0 ± 0.8	*/^†††^		0.224	rE = re = Re	0.590	0.181	0.510
M6	3.3 ± 1.0	3.3 ± 1.1	3.4 ± 0.8	3.2 ± 1.0	^†††^		0.704	rE = re = Re	0.272	0.695	0.482
M12	3.2 ± 0.9	3.5 ± 1.1	3.2 ± 0.6	3.0 ± 0.7	*/^†††^		0.649	rE = re = Re	0.375	0.570	0.182
**Hs-CRP**					**<0.001**	0.4765	0.682		0.323	0.254	0.858
D0	4.4 ± 3.8	3.7 ± 3.3	4.2 ± 3.3	5.2 ± 4.5							
D21	3.0 ± 3.6	2.3 ± 3.5	3.1 ± 2.9	3.8 ± 4.1	*		0.856	rE = re = Re	0.924	0.841	0.757
M3	3.5 ± 4.0	3.2 ± 4.7	2.7 ± 2.3	4.4 ± 4.3	^#/¥¥^		0.345	rE = re = Re	0.841	0.282	0.260
M6	2.7 ± 2.5	2.2 ± 1.8	2.9 ± 2.4	3.1 ± 3.0	***/^‡^		0.438	rE = re = Re	0.564	0.399	0.852
M12	2.5 ± 2.3	1.9 ± 2.3	2.8 ± 2.7	2.7 ± 1.9	***/^‡‡^		0.306	rE = re = Re	0.477	0.265	0.761
**Systolic Blood Pressure**					**<0.001**	0.5590	0.285		0.413	0.175	**0.031**
D0	139.3 ± 14.3	139.2 ± 12.1	137.1 ± 13.9	141.6 ± 16.6							
D21	130.0 ± 10.3	130.2 ± 8.7	129 ± 11.2	130.7 ± 11.2	***/^‡^/^#^/^¥¥¥^		0.351	rE = re = Re	0.532	0.362	0.739
M3	135.4 ± 12.5	134.4 ± 12	134.7 ± 12	137.0 ± 13.6	*/^†^		0.463	rE = re = Re	0.952	0.463	0.402
M6	135.8 ± 12.5	135.9 ± 9.5	134.7 ± 11.3	136.7 ± 16.1	*/^†^		0.401	rE = re = Re	0.583	0.411	0.753
M12	137.5 ± 12.1	134.0 ± 9.7	139.8 ± 11.7	139.1 ± 14.1	^†††^		0.103	rE >> Re	0.340	0.093	**0.006**
**Diastolic Blood Pressure**					**<0.001**	0.7698	0.831		0.342	0.906	0.297
D0	83.9 ± 9.8	84.5 ± 11.4	84.4 ± 8.6	82.9 ± 9.1							
D21	77.7 ± 9.7	77.5 ± 10.6	79.0 ± 11.5	76.6 ± 6.7	***/^‡‡‡^/^###^/^¥¥¥^		0.698	rE = re = Re	0.757	0.679	0.490
M3	83.9 ± 9.4	84.6 ± 10.1	84.2 ± 9.8	83.0 ± 8.3	^†††^		0.884	rE = re = Re	0.998	0.879	0.882
M6	84.3 ± 9.3	84.3 ± 8.8	84.6 ± 9.6	83.9 ± 9.7	^†††^		0.691	rE = re = Re	0.564	0.692	0.867
M12	83.8 ± 9.4	82.6 ± 7.6	85.7 ± 11.1	83.2 ± 9.5	^†††^		0.618	rE = re = Re	0.306	0.572	0.132

Re, rE and re: see methods for details. Bold for significant difference. D: day; M: month. HbA1c: glycated haemoglobin; Homa-IR: Homeostatic Model Assessment of Insulin Resistance, HDL: high-density lipoprotein, LDL: low-density lipoprotein, Hs-CRP: high-sensitivity c-reactive protein. BMI: Body Mass Index. *: *p* < 0.05 vs. D0; **: *p* < 0.01 vs. D0; ***: *p* < 0.001 vs. D0; ^†^: *p* < 0.05 vs. D20; ^††^: *p* < 0.01 vs. D20; ^†††^: *p* < 0.001 vs. D20; ^‡^: *p* < 0.05 vs. M3; ^‡‡^: *p* < 0.01 vs. M3; ^‡‡‡^: *p* < 0.001 vs. M3; ^#^: *p* < 0.05 vs. M6; ^##^: *p* < 0.01 vs. M6; ^###^: *p* < 0.001 vs. M6; ^¥^: *p* < 0.05 vs. M12; ^¥¥^: *p* < 0.01 vs. M12; ^¥¥¥^: *p* < 0.001 vs. M12. Re, rE and re: see methods for details.

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
