# Peer review of "Dietary Fibres and the Management of Obesity and Metabolic Syndrome: The RESOLVE Study"

_nutrients, 2020, doi:10.3390/nu12102911_

Round 1

Reviewer 1 Report

The authors have investigated the long-term maintenance of nutritional changes promoted during an intensive initial intervention to induce body weight loss. The authors have also evaluated the ability of these changes to predict long-term health outcomes was also examined.

This is an interesting article. The reported methods of analysis appear appropriate and the results do not appear to be over-interpreted. The authors have correctly highlighted strengths and limitations of study.

I found the paper to be overall well written and much of it to be well described.

SPECIFIC COMMENTS FOR REVISION

Introduction and Discussion - Results from the RESOLVE randomized controlled trial have been published in thirteen articles [18-30]. The authors should stress the novelty of the presented approach and reassure readers that the data have not been presented elsewhere.

Author Response

Dear Reviewer,

Thank you very much for your comment. We added the following sentence before the objective, within the introduction: “Despite the numerous articles published using data from the RESOLVE study, no article focused on the influence of dietary intake on body composition (especially fat tissue), and health outcomes”, as well as the following sentence at the beginning of the discussion: “No articles from the RESOLVE trial investigated to date the influence of nutrition aspects”.

We hope our work will be considered favorably and look forward to hearing from you.

Sincerely yours,

                                                                           Frédéric Dutheil

Reviewer 2 Report

Overall a very well-written tight paper. I would love to see a little more written about the relationship of adherence to the prescribed diet with the exercise. Did the authors see any fall off with exercise compliance as they may have with diet compliance and if so, any relationships between the outcomes that can be connected with exercise as well as diet type? The authors did a great job overall and would like to see this aspect covered a little more in the results and discussion sections of the paper.

Author Response

Dear Reviewer,

Thank you very much for your comment. We added the following sentences within the methods: “All outcomes were measured at baseline (D0), at 21 days (D21) i.e. at the end of the residential program, and during the at-home follow-up at 3 months (M3), 6 months (M6) and 12 months (M12), except for food questionnaires and number of training sessions per week that were asked every month. They were used to calculate compliance to nutrition (score from 0 to 12 i.e. 12 = 100% for food questionnaires returned) and to exercise (score from 0 to 4, i.e. 4 = 100%  for the number of training sessions undertaken per week). The overall compliance score was the mean of these two scores (nutrition and physical activity).” We added the following sentences within the results: “We included 100 volunteers (59.4±5.1 years old, 44% men) who underwent randomization; 87 completed the whole intervention (Figure 1), without difference between participants who withdraw from the study for sociodemographic. The mean overall compliance score to both diet and exercise was 52.4 ± 22.4% without difference between groups. There is a significant correlation (r=0.72) between compliance to diet and compliance to exercise (p<0.0001).” We also added the following sentences within the discussion: “The mean compliance to diet and exercise guidelines was adequate in the present study. However, we want to acknowledge that maintaining a such level of high compliance was costly and therefore the study may lack of generalizability as a public health strategy within the community.”

We hope our work will be considered favorably and look forward to hearing from you.

Sincerely yours,

                                                                           Frédéric Dutheil
